  

# FAM35A associates with REV7 and modulates DNA damage responses of normal and BRCA1-defective cells

Junya Tomida[1],* , Kei-ichi Takata[1], Sarita Bhetawal[1], Maria D Person[2], Hsueh-Ping Chao[1], Dean G Tang[3] & Richard D Wood[1],**

## Abstract

To exploit vulnerabilities of tumors, it is urgent to identify associated defects in genome maintenance. One unsolved problem is the mechanism of regulation of DNA double-strand break repair by REV7 in complex with 53BP1 and RIF1, and its influence on repair pathway choice between homologous recombination and non-homologous end-joining. We searched for REV7-associated factors in human cells and found FAM35A, a previously unstudied protein with an unstructured N-terminal region and a C-terminal region harboring three OB-fold domains similar to single-stranded DNA-binding protein RPA, as novel interactor of REV7/RIF1/53BP1. FAM35A re-localized in damaged cell nuclei, and its knockdown caused sensitivity to DNA-damaging agents. In a BRCA1-mutant cell line, however, depletion of FAM35A increased resistance to camptothecin, suggesting that FAM35A participates in processing of DNA ends to allow more efficient DNA repair. We found FAM35A absent in one widely used BRCA1-mutant cancer cell line (HCC1937) with anomalous resistance to PARP inhibitors. A survey of FAM35A alterations revealed that the gene is altered at the highest frequency in prostate cancers (up to 13%) and significantly less expressed in metastatic cases, revealing promise for FAM35A as a therapeutically relevant cancer marker.

**Keywords** camptothecin; cisplatin; DNA repair; olaparib; prostate cancer
**Subject Categories** Cancer; DNA Replication, Repair & Recombination
**The EMBO Journal (2018) 37: e99543**

## Introduction

REV7 is a multifunctional protein encoded by the *MAD2L2* gene in human cells. REV7 acts as an interaction module in several cellular pathways. One of its functions is as a component of DNA polymerase ζ, where it serves as bridge between the Pol ζ catalytic subunit REV3L and the REV1 protein. A dimer of REV7 binds to two adjacent sites in REV3L by grasping a peptide of REV3L with a "safety-belt" loop (Hara *et al*, 2010; Tomida *et al*, 2015). REV7 protein is an order of magnitude more abundant than REV3L (Tomida *et al*, 2015) and has additional functions and protein partners including chromatin-associated and post-translational modification proteins (Medendorp *et al*, 2009; Vermeulen *et al*, 2010; Itoh *et al*, 2011; Listovsky & Sale, 2013; Pirouz *et al*, 2013; Hara *et al*, 2017). Complete ablation of REV7 gives rise to mice with defects in primary germ cells (Pirouz *et al*, 2013; Watanabe *et al*, 2013). Recently, studies uncovered a function of REV7 as a DNA resection inhibitor, limiting genomic repair by an unknown mechanism (Boersma *et al*, 2015; Xu *et al*, 2015). Although BRCA1-mutant cells are defective in homologous recombination, these studies found that one mode to partially restore recombination activity is by inactivation of REV7. It was proposed that REV7, together with 53BP1 and RIF1, inhibits 5′ DNA end resection to promote non-homologous end-joining at the expense of homologous recombination.

To investigate novel functions and pathways involving REV7, we identified proteins associated with REV7 *in vivo*. We report here an analysis of a previously uncharacterized REV7-interacting protein, FAM35A. We discovered that FAM35A is a novel factor that modulates the DNA damage sensitivity of normal and BRCA1-defective cells. Our analysis reveals that the C-terminal half of FAM35A contains three OB-fold domains similar to those in the single-stranded DNA-binding protein RPA large subunit. FAM35A has a disordered N-terminal portion, containing sites of DNA damage-dependent post-translational modification. Moreover, the *FAM35A* gene is deleted at an unusually high rate in prostate cancers, and in cells from at least one well-studied BRCA1-defective breast cancer case. FAM35A is more weakly expressed in metastatic prostate cancers, suggesting it as an important marker for outcome and therapeutic decisions.

1 Department of Epigenetics & Molecular Carcinogenesis, The University of Texas MD Anderson Cancer Center, Smithville, TX, USA
2 Proteomics Facility, University of Texas at Austin, Austin, TX, USA
3 Department of Pharmacology and Therapeutics, Roswell Park Cancer Institute, Buffalo, NY, USA
*Corresponding author. Tel: +1 512 237 6433; E-mail: jtomida@mdanderson.org
**Corresponding author. Tel: +1 512 237 9431; E-mail: rwood@mdanderson.org

# Results and Discussion

### FAM35A interacts with REV7, 53BP1, and RIF1 *in vivo*

To isolate proteins associated with REV7, we engineered HeLa S3 cells that stably express REV7 with a C-terminal FLAG–HA epitope tag (REV7-FH). REV7-FH was sequentially immunoprecipitated from nuclear extract using FLAG and HA antibody beads

(Ikura *et al*, 2007). This purified complex was separated by gradient gel electrophoresis, and associated proteins from gel sections were identified by LC-MS/MS. We confirmed association with previously identified REV7-binding proteins including GLP (Nakatani & Ogryzko, 2003; Pirouz *et al*, 2013), G9A (Pirouz *et al*, 2013), CAMP (Itoh *et al*, 2011), GTF2I (Fattah *et al*, 2014), POGZ (Vermeulen *et al*, 2010), and HP1α (Vermeulen *et al*, 2010) (Fig 1A and Table 1). The highest-ranking previously unstudied

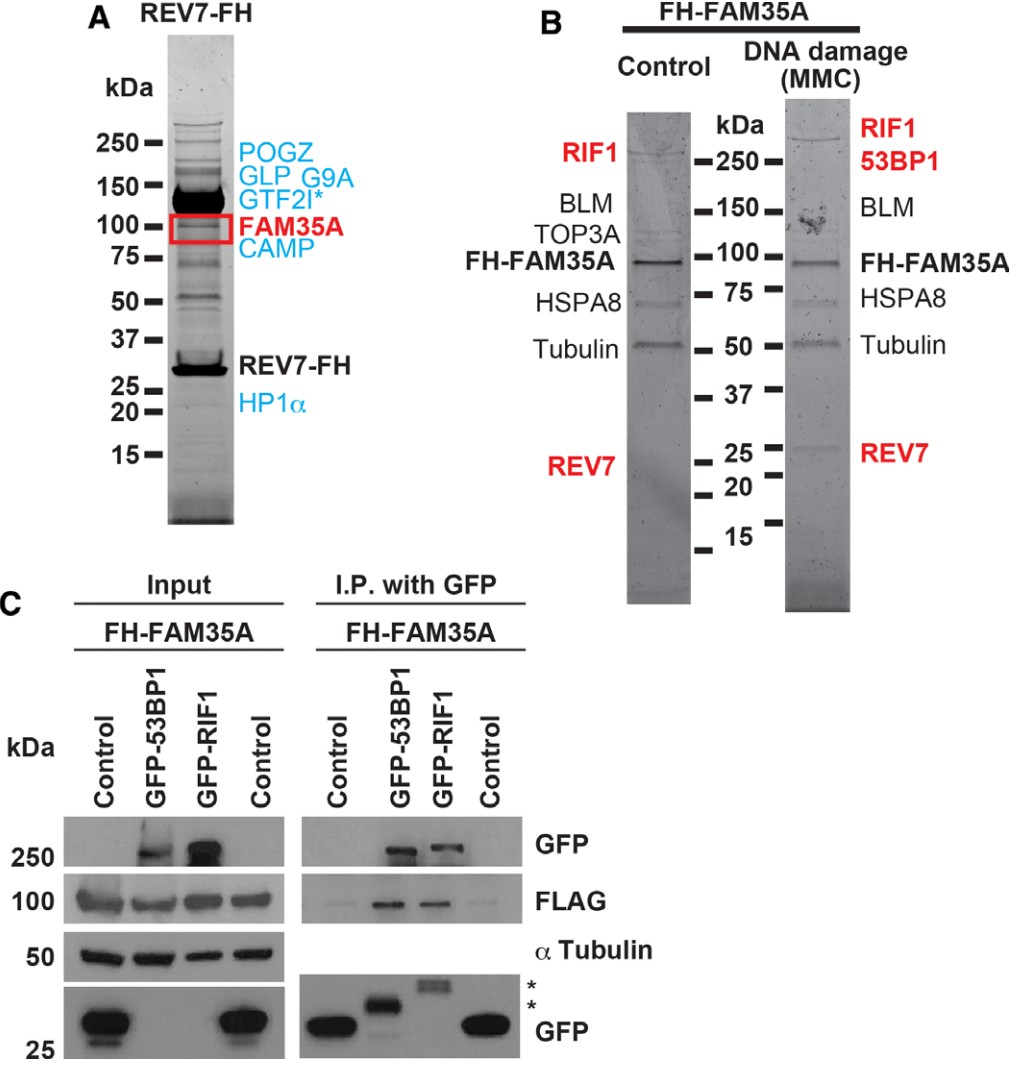

**Figure 1. Identification of REV7- or FAM35A-associated proteins.**

Protein complexes were sequentially immunoprecipitated (using FLAG and HA antibody beads) from nuclear extracts of HeLa S3 cell lines stably expressing C-terminally FLAG–HA-tagged REV7 (REV7-FH) or (N-terminally FLAG–HA-tagged-FAM35A (FH-FAM35A).

A, B   REV7 complex (A) and FAM35A complex (B); associated proteins were identified by mass spectrometry. The FH-FAM35A complex was purified from HeLa S3 nuclear extracts after 18 h exposure to MMC (100 ng/ml). Proteins labeled in blue are previously published REV7-binding partners. Proteins labeled in red are involved in end-joining pathways of DSB repair. 4–20% gradient gels were stained with SYPRO Ruby.

C      FH-FAM35A was co-transfected into human 293T cells with GFP empty vector (control), GFP-53BP1 or GFP-RIF1. Forty-eight hours after transfection, cell lysates were made and used for immunoprecipitation with GFP antibody beads. After electrophoretic transfer of proteins, the membrane was cut into three sections to separate proteins > 250 kDa (GFP as GFP-53BP1 and GFP-RIF1), 37–250 kDa (α-tubulin), and < 37 kDa (GFP as control) and immunoblotted with the indicated antibodies. Results for the input and immunoprecipitation (IP) product after gel electrophoresis are shown. The asterisk (*) in the IP lane marks degraded or truncated forms of 53BP1 and RIF1.

Source data are available online for this figure.

**Table 1.  Identification of FAM35A with previously reported proteins in a REV7-associated complex.**

| Protein | Accession number (UniProtKB) | Molecular weight | Spectral counts | Unique peptides |
|---|---|---|---|---|
| REV7 | Q9UI95\|MD2L2_HUMAN | 24 kDa | 453 | 15 |
| FAM35A | Q86V20-2\|FA35A_HUMAN | 92 kDa | 74 | 23 |
| GLP | Q9H9B1\|EHMT1_HUMAN | 141 kDa | 9 | 6 |
| G9A | Q96KQ7\|EHMT2_HUMAN | 135 kDa | 7 | 5 |
| CAMP | Q96JM3\|ZN828_HUMAN | 89 kDa | 127 | 29 |
| GTF2I | B4DH52\|B4DH52_HUMAN | 112 kDa | 1780 | 88 |
| POGZ | Q7Z3K3\|POGZ_HUMAN | 154 kDa | 147 | 31 |
| HP1α | P45973\|CBX5_HUMAN | 22 kDa | 8 | 3 |

The immunoprecipitated sample was separated on a denaturing polyacrylamide gel, and proteins from gel sections of approximately equal size were identified by LC-MS/MS. FAM35A was a top specific hit, together with previously identified REV7-associated proteins. GTF2I (TFII-I) is likely a non-significant association as it is an abundant protein frequently found in control experiments with agarose supports and Flag-His tags (www.crapome.org), but it is included here for reference, as it was previously reported to interact with REV7 (Fattah *et al*, 2014).

**Table 2.  DNA repair proteins identified in the FAM35A complex.**

| Protein | Accession number (UniProtKB) | Molecular weight | Spectral counts | Unique peptides |
|---|---|---|---|---|
| FAM35A | Q86V20-2\|FA35A_HUMAN | 92 kDa | 170 | 37 |
| REV7 | Q9UI95\|MD2L2_HUMAN | 24 kDa | 5 | 2 |
| RIF1 | Q5UIP0-2\|RIF1_HUMAN | 272 kDa | 6 | 4 |
| BLM | H0YNU5\|H0YNU5_HUMAN | 144 kDa | 6 | 4 |
| TOP3A | Q13472\|TOP3A_HUMAN | 112 kDa | 5 | 3 |

Proteins immunoprecipitating with exogenously expressed FAM35A were separated on a denaturing polyacrylamide gel, and proteins from gel sections of approximately equal size were identified by LC-MS/MS. Table shows the significant DNA repair-related proteins that were detected.

**Table 3.  DNA repair proteins identified in the FAM35A complex after MMC exposure.**

| Protein | Accession number (UniProtKB) | Molecular weight | Spectral counts | Unique peptides |
|---|---|---|---|---|
| FAM35A | Q86V20-2\|FA35A_HUMAN | 92 kDa | 623 | 69 |
| RIF1 | Q5UIP0-2\|RIF1_HUMAN | 272 kDa | 315 | 112 |
| REV7 | Q9UI95\|MD2L2_HUMAN | 24 kDa | 21 | 4 |
| Ku80 | P13010\|XRCC5_HUMAN | 83 kDa | 6 | 4 |
| BLM | H0YNU5\|H0YNU5_HUMAN | 144 kDa | 6 | 4 |
| 53BP1 | Q12888-2\|TP53B_HUMAN | 214 kDa | 3 | 3 |
| Ku70 | P12956\|XRCC6_HUMAN | 70 kDa | 11 | 7 |

Proteins from gel sections were identified by LC-MS/MS. The gel sections were approximately equal size with the exception of a wider segment for the FAM35A bait. The top associated hits were RIF1 and REV7. All other known DNA repair-related proteins that were detected are shown. Ku70 and Ku80 may be non-significant associations as they are found frequently in control experiments with agarose supports and Flag-His tags (www.crapome.org), but they are included here as they were not detected with significance in the complex from non-damaged cells (Table 2).

spectrometry (Fig 1B). DNA repair proteins associating with FAM35A included REV7, RIF1, BLM, and TOP3A (Table 2), with relatively more RIF1 peptides and 53BP1 identified following MMC exposure (Table 3).

Although REV7 is known to cooperate functionally with 53BP1 to limit resection at DNA breaks, REV7 was not detected in 53BP1 immunocomplexes, and it is unknown how REV7 connects with 53BP1 *in vivo* (Xu *et al*, 2015). To verify an association of FAM35A with 53BP1 and the additional DNA end resection control factor RIF1, FH-FAM35A was co-transfected with GFP-RIF1 or GFP-53BP1 and expressed in 293T cells. All proteins were expressed at the predicted molecular weights (Fig 1C, "input" lanes). Following immunoprecipitation with GFP antibody beads, FAM35A co-immunoprecipitated with recombinant RIF1 or 53BP1 (but not with the control vector, Fig 1C). These interactions suggest that FAM35A may functionally bridge 53BP1 and REV7 in human cells, directly or via interactions with other proteins.

### FAM35A is an OB-fold protein that changes localization following DNA damage

We found *FAM35A* orthologs are present in vertebrate genomes, but not in invertebrates or plants. Multiple protein isoforms arising from alternative splicing are annotated in genomics databases for human (UniProt accession number Q86V20) and mouse FAM35A. Isoforms 1 and 2 are the most common, encoding 904 and 835 amino acid proteins, respectively. They arise by differential splicing of one in-frame exon (Fig 2A). Both mRNA isoforms of FAM35A are ubiquitously expressed in different cell and tissue types (www.gtexportal.org).

association was with the uncharacterized FAM35A protein (Fig 1A).

To validate the association, a reciprocal experiment was performed by constructing a HeLa S3 cell line stably expressing FAM35A with an N-terminal FLAG–HA tag (FH-FAM35A). Cells were exposed to mitomycin C (MMC, 100 ng/ml) for 18 h or mock-exposed. Following sequential immunoprecipitation with FLAG and HA antibody beads, proteins were separated and identified by mass

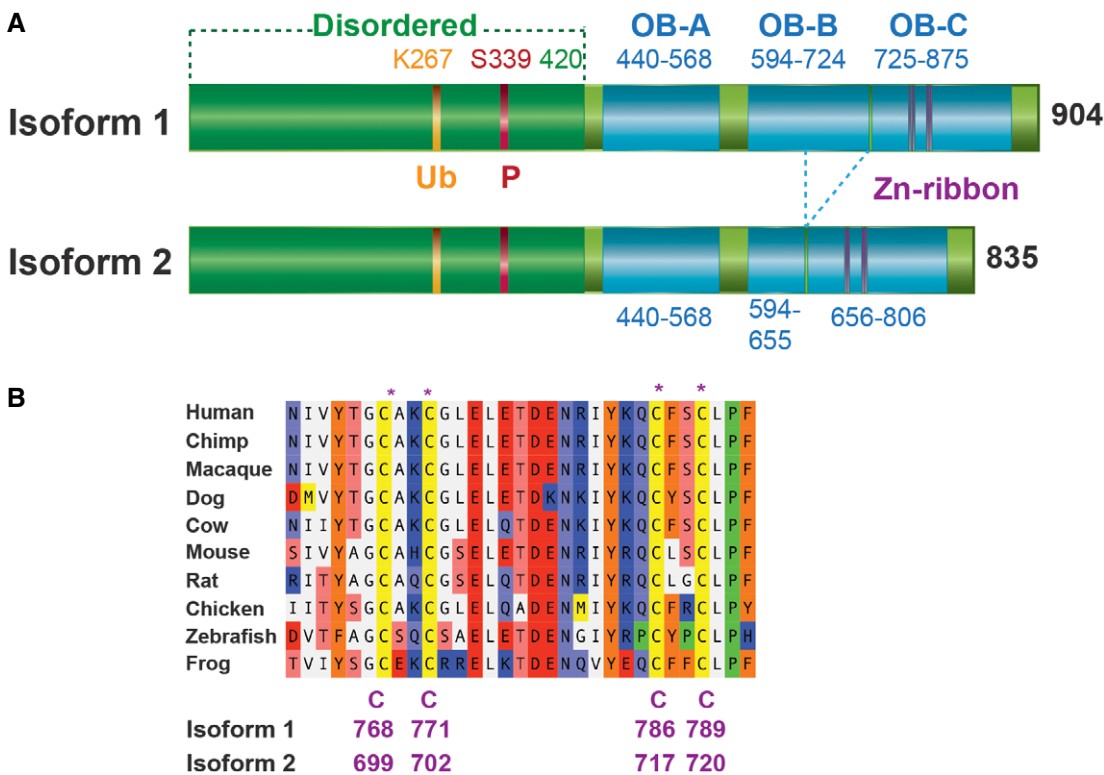

**Figure 2. FAM35A is an OB-fold protein with an N-terminal disordered region.**

A  Domain schematic of human FAM35A derived from sequence prediction modeling. An N-terminal disordered region includes post-translational modification sites. Locations of the three OB-fold domains A, B, and C are shown, with a Zn-ribbon containing conserved Cys residues. One exon is absent in isoform 2 compared to isoform 1, deleting 69 aa from OB domain B.

B  Multi-species alignment of a segment of FAM35A protein in the predicted Zn-ribbon. The four Zn-coordinating Cys residues (CxxC, CxxC), homologous to those in human RPA1, are evolutionarily conserved.

BLAST searches for sequence homologs did not reveal significant primary sequence similarity to gene products other than FAM35A. We therefore analyzed the FAM35A protein sequence using structure prediction servers based on PSI-BLAST. The N-terminal half of the protein is predicted to be disordered up until about residue 420 (Fig 2A), and this region is likely to interact with other proteins, as found commonly for disordered regions of polypeptides (Receveur-Brechot *et al*, 2006). The N-terminal region contains previously identified post-translational modification sites, including a conserved ubiquitin modification (Kim *et al*, 2011) and a conserved SQ site (Matsuoka *et al*, 2007) in which residue S339 is phosphorylated after exposure to ionizing radiation or UV radiation (Matsuoka *et al*, 2007). With high significance, the C-terminal portion of FAM35A is predicted to contain three OB-fold domains structurally homologous to those in the 70 kDa subunit (RPA1) of the single-stranded DNA-binding protein RPA (Figs 2A and EV1). The three OB folds are similar to DNA-binding domain folds A, B, and C of RPA (Bochkareva *et al*, 2002; Fan & Pavletich, 2012). OB-fold domain C is predicted to include four conserved cysteine residues (Fig 2B) at the core of a zinc-binding ribbon, homologous to a loop in the same position in RPA (Fig EV2). Together, the OB folds and Zn-ribbon form the elements of DNA-binding and orientation-enabling RPA1 to simultaneously bind to protein partners and to single-stranded DNA in

an 8–10 nt binding mode (Lin *et al*, 1998; Bochkareva *et al*, 2002; Arunkumar *et al*, 2003). The region including the 4-Cys Zn-ribbon identifies the domain of unknown function (PF15793) that is conserved in FAM35A homologs as annotated by the Pfam database. C-terminal helices are predicted present following OB domains A and B, in positions corresponding to helices involved in multimerization of the OB folds in RPA subunits (Bochkareva *et al*, 2002; Fan & Pavletich, 2012; Fig EV2).

We constructed a human U2OS cell line stably expressing GFP-FAM35A. Cells were exposed to MMC (100 ng/ml, 24 h) or mock-exposed and then fixed and stained with DAPI and anti-GFP. In cells exposed to DNA-damaging agent, GFP-FAM35A was concentrated into foci in the nucleus (Fig 3A), suggesting a direct involvement in DNA repair.

**FAM35A depletion sensitizes cell lines to DNA damage**

The human *FAM35A* gene is located on chromosome 10q23.2. Three pseudogenes are also present in the human genome, two of them on 10q22 (Fig 3B) with high (> 98%) sequence identity to *FAM35A*. Precise nuclease-mediated knockout of human *FAM35A* is therefore challenging, as simultaneous targeting of pseudogenes would likely cause chromosome rearrangements and deletion. siRNA was used to acutely deplete FAM35A from human HEK293 cells and

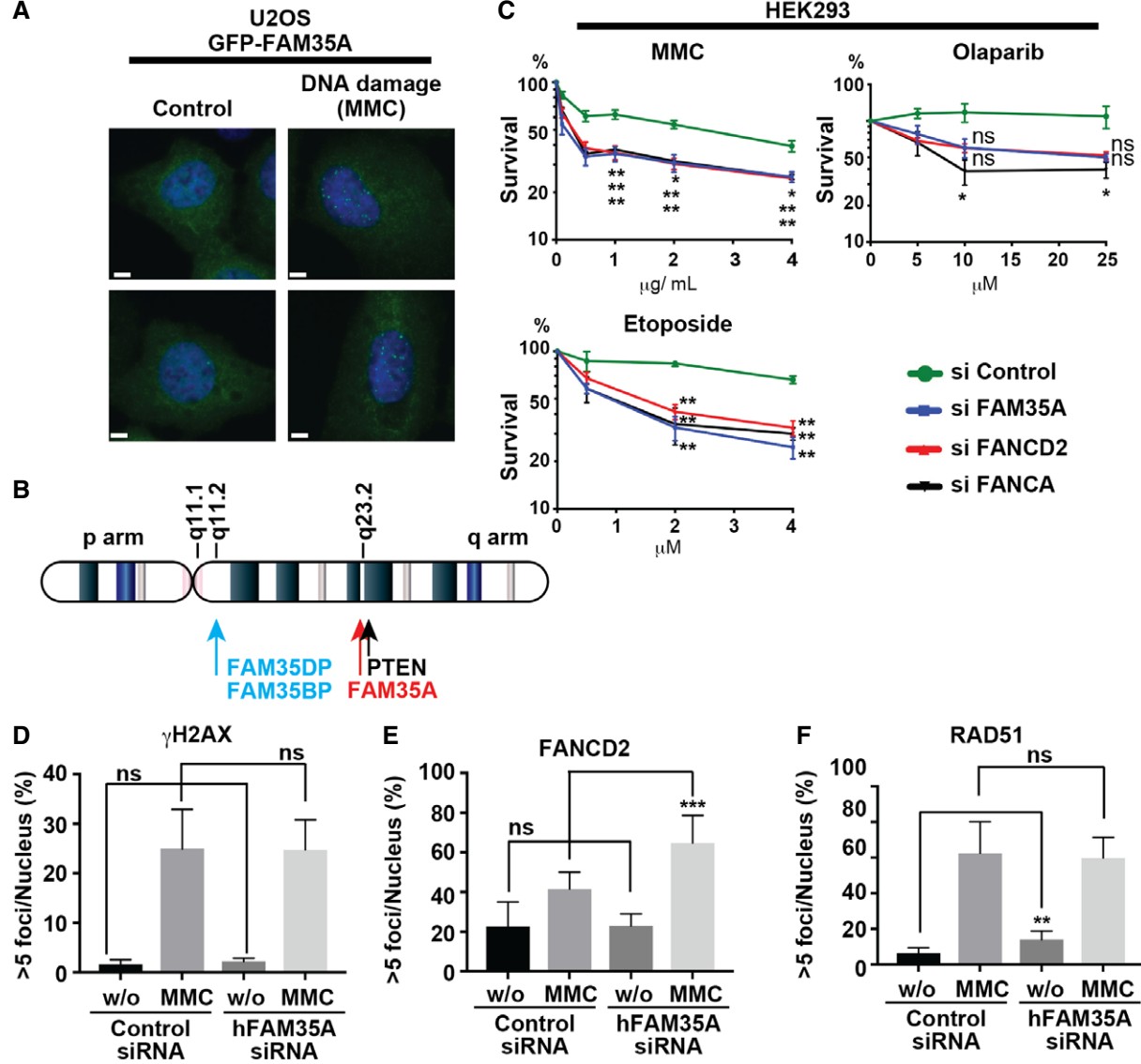

**Figure 3. FAM35A is a DNA damage response gene.**

A    GFP-FAM35A forms nuclear foci upon DNA damage. A U2OS cell line stably expressing GFP-FAM35A was exposed to MMC (100 ng/ml, 24 h) (right) or to mock treatment (left). The following day, cells were fixed and stained for DAPI and anti-GFP. Scale bars: 6 μm.

B    Two pseudogenes (*FAM35DP* and *FAM35BP*) with > 98% identity to *FAM35A* are located on chromosome 10q22. Both *FAM35DP* and *FAM35BP* are present in genomes of apes and old-world monkeys, but not in other mammalian genomes. By inference, these pseudogenes arose by whole gene duplication in the common ancestor of the catarrhines about 25–30 million years ago. A third pseudogene (not shown) is *FAM35CP*, an inactive spliced product of reverse transcription (> 95% identity) that was integrated into an intron of the galactosylceramidase (*GALC*) gene on chromosome 14q31.3. *FAM35CP* is present in apes but not old-world monkeys, indicating a more recent evolutionary origin.

C    Acute depletion of FAM35A causes hypersensitivity to several DNA-damaging agents but not to olaparib. The survival of HEK293 cells, FAM35A acutely depleted and control, was monitored following exposure to MMC, etoposide, and olaparib. siControl (circle symbol, green line). siFAM35A (square symbol, blue line). siFANCD2 (triangle symbol, red line). siFANCA (triangle symbol, black line). siRNA-treated cells were plated and exposed to indicate dose of agent for 48 h. Cellular viability was measured 48 h later. Data represent mean ± SEM. n = 3. *P < 0.05 and **P < 0.01 by unpaired *t*-test.

D–F    Quantification analysis of > 5 nuclear γH2AX foci (D), > 5 FANCD2 foci (E), and > 5 RAD51 foci (F) in HEK293 cells acutely depleted for FAM35A. Cells with > 5 foci/nucleus were counted after MMC (100 ng/ml, 24 h) or control treatment. Data shown are means ± SE of more than 250 nuclei from two independent experiments. ns: not significant, **P < 0.01 and ***P < 0.001 by unpaired *t*-test.

Source data are available online for this figure.

investigate its role in DNA repair. FAM35A-depleted HEK293 cultures were hypersensitive to MMC and etoposide, with sensitivity comparable to that conferred by depletion of Fanconi anemia *FANCA* and *FANCD2* gene expression (Fig 3C). In HEK293 cells, *FAM35A*-siRNA did not sensitize to the PARP inhibitor olaparib (Fig 3C), suggesting that homologous recombination repair is still

active in the absence of FAM35A, as it is in cells with suppressed REV7 activity (Boersma *et al*, 2015; Xu *et al*, 2015).

We investigated the impact of acute depletion of FAM35A on other markers of DNA damage responses in HEK293 cells. Cells with > 5 γH2AX foci, > 5 FANCD2 foci, or > 5 RAD51 foci were quantified after MMC exposure (100 ng/ml, 24 h) and in control cells. MMC exposure induced cellular γH2AX foci as expected; there was no significant change in this pattern in cells depleted for FAM35A, indicating intact signaling leading to γH2AX formation (Fig 3D). The Fanconi anemia signaling pathway (FANCD2 foci) was also functional in FAM35A-depleted cells, showing about 1.5-fold more foci after MMC exposure (Fig 3E). RAD51 foci, a readout of homologous recombination pairing, were formed in FAM35A-defective

cells, with a twofold elevated frequency in non-damaged cells (Fig 3F).

Depletion of REV7 reduces the efficiency of non-homologous end-joining (NHEJ; Boersma *et al*, 2015; Xu *et al*, 2015) by promoting resection that channels repair of DNA double-strand breaks into homologous recombination and other pathways. Because of the association between FAM35A and the resection control factors REV7, RIF1, and 53BP1 (Fig 1A–C), we investigated whether FAM35A depletion affects end joining, using a plasmid integration assay (Boersma *et al*, 2015). REV7, 53BP1, and RIF1 depletion decreased integration ratio in this assay. We confirmed efficient knockdown of *FAM35A* mRNA and protein in 293T cells using qPCR (Fig 4A) and immunoblot analysis (Fig 4B). The plasmid integration

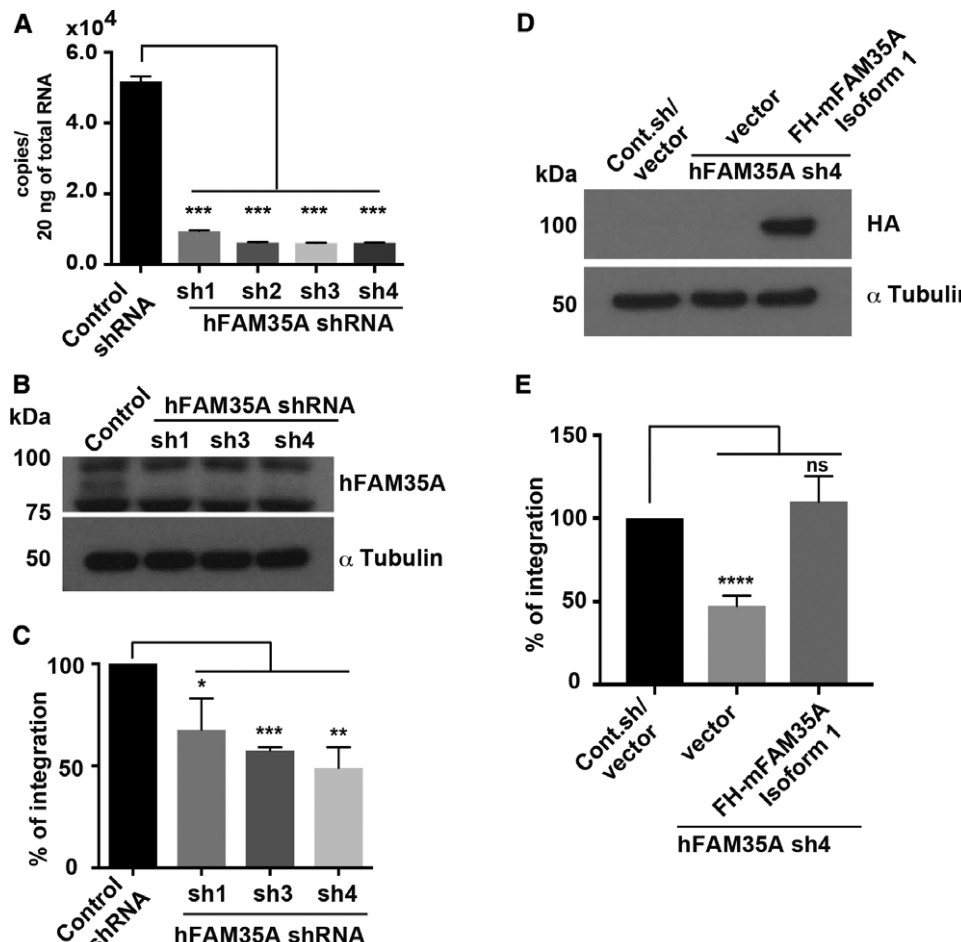

**Figure 4.  Suppression of FAM35A reduces random DNA integration by non-homologous end-joining.**

A, B   Depletion of endogenous FAM35A mRNA (A) and protein (B) from 293T cell lines carrying shRNA to FAM35A. Cell lysates were made from 293T cell lines stably depleted for FAM35A and controls. After electrophoretic transfer of proteins, the membrane was immunoblotted with anti-FAM35A or anti-α-tubulin.

C   Plasmid integration assay. pDsRed-Monomer plasmid with antibiotic resistance (Hygromycin B) was linearized by restriction enzyme. The linearized plasmid was transfected into stably FAM35A-depleted 293T cell lines and control. Colonies were counted after antibiotic selection (Hygromycin B 300 μg/ml).

D   Expression of FAM35A isoform 1 (mouse) in human 293T cells with stable suppression of human FAM35A. The immunoblot of cell lysates used an anti-HA antibody to recognize epitope-tagged mouse FAM35A.

E   Plasmid integration frequency is restored by expression of FAM35A in cells.

Data information: Data represent mean ± SEM. $n = 4$. *$P < 0.05$, **$P < 0.01$, ***$P < 0.001$ and ****$P < 0.0001$ by unpaired *t*-test.
Source data are available online for this figure.

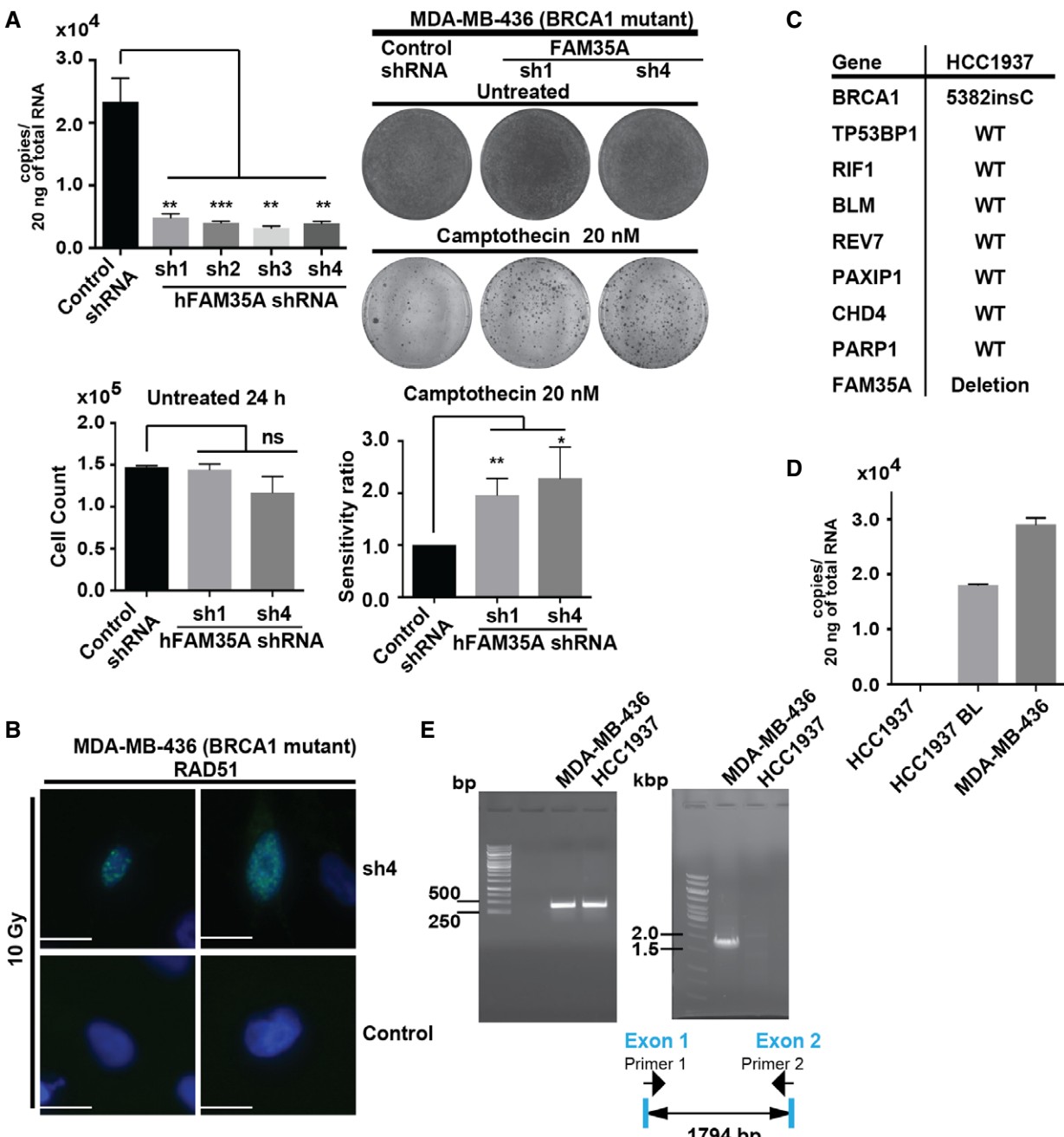

**Figure 5.** **FAM35A depletion/KO in BRCA1-deficient cells enhances markers of homologous recombination.**

A   Quantification of depletion of endogenous *FAM35A* mRNA and protein from 293T cell lines carrying shRNA to *FAM35A*. Colony-forming assay: MDA-MB-436 cells infected with non-targeting control or *FAM35A* shRNAs were treated for 24 h with 20 nM camptothecin, with medium changed and incubated until colonies appear; cells were then fixed and stained. Top row is untreated; second row is treated with camptothecin. Colony and cell numbers were counted. Bottom left is the number of cells 24 h after seeding $1.5 \times 10^5$ cells. Bottom right is the colony count. Data represent mean ± SEM. *n* = 3. ns: not significant, *P < 0.05, **P < 0.01 and ***P < 0.001 by unpaired *t*-test.

B   FAM35A-depleted BRCA1-mutant cells form nuclear foci of RAD51 following DNA damage. MDA-MB-436 cells infected with non-targeting control (second row) or *FAM35A* shRNAs (top row) were exposed to X-rays (10 Gy). Cells were fixed after 6 h and stained with DAPI and anti-RAD51. Scale bars: 10 μm.

C   Genes related to PARP inhibitor resistance in *BRCA1*-mutant cells and alterations of these genes in the HCC1937 cell line according to the CCLE.

D   Quantification of endogenous *FAM35A* mRNA expression. Endogenous *FAM35A* mRNA from HCC1937 (*BRCA1* mutant), HCC1937BL, and MDA-MB-436 (*BRCA1* mutant) cell lines was quantified using qPCR. Data represent mean ± SEM. *n* = 3.

E   Detection of FAM35A exons 1 and 2 deletion in genomic DNA using PCR. Deletion of FAM35A exons 1 and 2 is confirmed in the HCC1937 cell line. In HCC1937 BL, endogenous FAM35A mRNA expression level was detectable. PCR primers for amplification of genomic DNA were designed in exon 1 (forward; primer 1) and exon 2 (reverse; primer 2). The predicted PCR product size is 1,794 bp.

Source data are available online for this figure.

ratio decreased significantly after FAM35A depletion (Fig 4C), suggesting that FAM35A is involved in modulating double-strand break repair pathway choice. Expression of FAM35A isoform 1 in 293T cells (Fig 4D) restored NHEJ to normal levels (Fig 4E). This is consistent with increased resection in the absence of FAM35A, causing NHEJ to be less effective, which may account for the increased sensitivity of FAM35A-depleted cells to MMC and etoposide.

### FAM35A deficiency in BRCA1-deficient cells and resistance to camptothecin and PARP inhibitors

In BRCA1-mutant cells, REV7 depletion restores homologous recombination (HR) by restoring 5′ end resection (Boersma et al, 2015; Xu et al, 2015). We therefore hypothesized that FAM35A depletion from BRCA1-mutant cells would increase resistance to a DNA strand-breaking agent. We engineered a BRCA1-mutant cell line (MDA-MB-436) expressing FAM35A shRNA. Efficient knockdown was verified by qPCR (Fig 5A). The FAM35A-depleted BRCA1-mutant cells and controls were assayed for sensitivity to camptothecin using a colony-forming assay. FAM35A depletion from the BRCA1-mutant cell line significantly alleviated the sensitivity to camptothecin (Fig 5A). Following exposure to ionizing radiation, a BRCA1-mutant cell line transfected with control siRNA did not form RAD51 foci, as expected (Johnson et al, 2013). However, the BRCA1-mutant cells formed damage-dependent nuclear RAD51 foci following FAM35A depletion, suggesting that 5′ end resection was more active in the absence of FAM35A (Fig 5B).

We investigated a further widely used cell line, HCC1937, which has a known inactivating mutation in BRCA1 (Fig 5C). Multiple studies have noted that HCC1937 is anomalously resistant to PARP inhibitors in comparison with other BRCA1-mutant cell lines, for

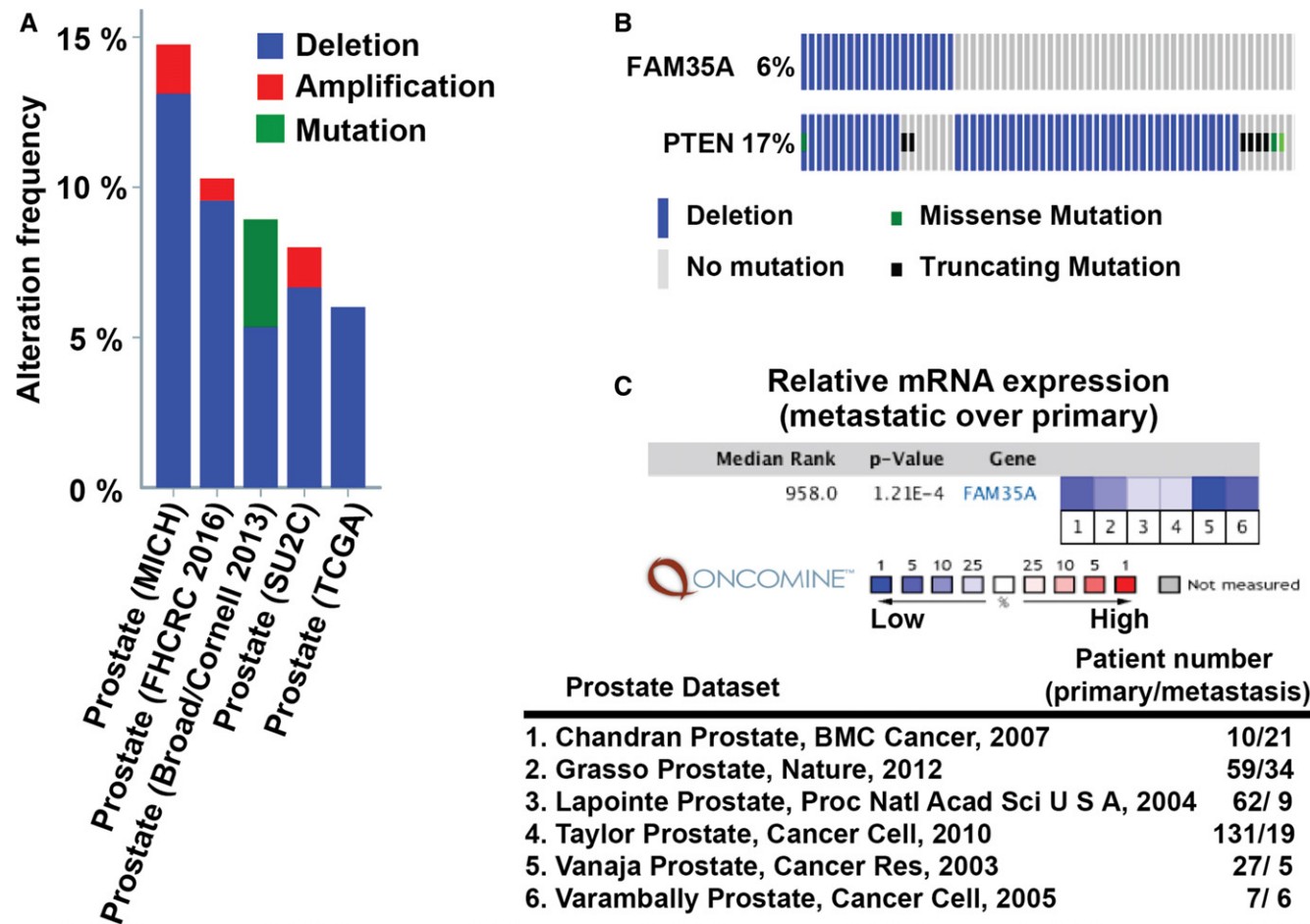

**Figure 6.  FAM35A is a candidate tumor/metastasis suppressor gene in prostate cancer.**

A  Prevalence of FAM35A deletion in prostate cancer (PCa) genomes analyzed via the cBioPortal. FAM35A gene is depleted in ~6–13% of PCa, in the studies indicated.

B  FAM35A deletion is sometimes independent of alteration in PTEN. Comparison of FAM35A and PTEN alterations in PCa according to TCGA data (333 samples, Cancer Genome Atlas Network, 2015) from the cBioPortal. Blue bar, homozygous deletion. Gray bar, no mutation. Green square, missense mutation. Black square, truncating mutation.

C  FAM35A mRNA expression in primary vs. metastatic tumor according to Oncomine. The P-value for a gene is its P-value for the median-ranked analysis. The enumerated studies are Chandran Prostate (Chandran et al, 2007), Grasso Prostate (Grasso et al, 2012), Lapointe Prostate (Lapointe et al, 2004), Taylor Prostate (Taylor et al, 2010), Vanaja Prostate (Vanaja et al, 2003), and Varambally Prostate (Varambally et al, 2005).

unknown reasons (Drew *et al*, 2011; Lehmann *et al*, 2011; Pierce *et al*, 2013). In HCC1937, RAD51 foci (a marker of DNA end resection) are generated after DNA damage exposure (Zhang *et al*, 2004; Hill *et al*, 2014), and homologous recombination is partially active despite BRCA1 mutation (Zhang *et al*, 2004). We found that *FAM35A* mRNA expression is absent in HCC1937 cells (Fig 5D). FAM35A expression was present in HCC1937BL, an EBV-immortalized lymphoblastoid cell line from the same patient (Fig 5D). HCC1937BL is heterozygous for the BRCA1 mutation, indicating loss of heterozygosity for both FAM35A and BRCA1 in the HCC1937 mammary cancer patient.

Genomic analysis of DNA from HCC1937 and MDA-MB-436 cells showed that *FAM35A* DNA flanking exons 1 and 2 were not present in HCC1937 cells (Fig 5E). This is consistent with a deep deletion at the *FAM35A* locus, notated by the Cancer Cell Line Encyclopedia (CCLE, Broad Institute) in HCC1937 cells (Fig EV3A). Reported genes related to PARP inhibitor resistance in BRCA1-mutant cells include *53BP1, MAD2L2/REV7, RIF1, BLM, CHD4,* and *PTIP*

(Bunting *et al*, 2010; Boersma *et al*, 2015; Xu *et al*, 2015; Patel *et al*, 2017). None are deleted in HCC1937 (Fig 5C). DNA double-strand break repair factors ATM (Boersma *et al*, 2015), NBS1 (Boersma *et al*, 2015), and RNF8 (Boersma *et al*, 2015) are also preserved in HCC1937 according to the CCLE.

These data suggest an involvement of FAM35A in resection inhibition in parallel with REV7, 53BP1, and RIF1 (Boersma *et al*, 2015; Xu *et al*, 2015). Like 53BP1-depleted BRCA1 mutants (Bunting *et al*, 2010; Densham *et al*, 2016), FAM35A-depleted BRCA1-mutant cells acquire camptothecin resistance. As HCC1937 cells can form 53BP1 foci and Mre11 foci (Zhang *et al*, 2004) following DNA damage (Hill *et al*, 2014), FAM35A appears to operate downstream of these factors and may serve as a link between 53BP1 and REV7. It is notable that the HCC1937 cell line relies on Pol θ-dependent alternative end-joining for survival, perhaps unusually so for a BRCA1-defective cell line (Mateos-Gomez *et al*, 2015). A likely explanation is that the excessive DNA end resection activity arising from the FAM35A defect channels a substantial fraction of breaks into repair

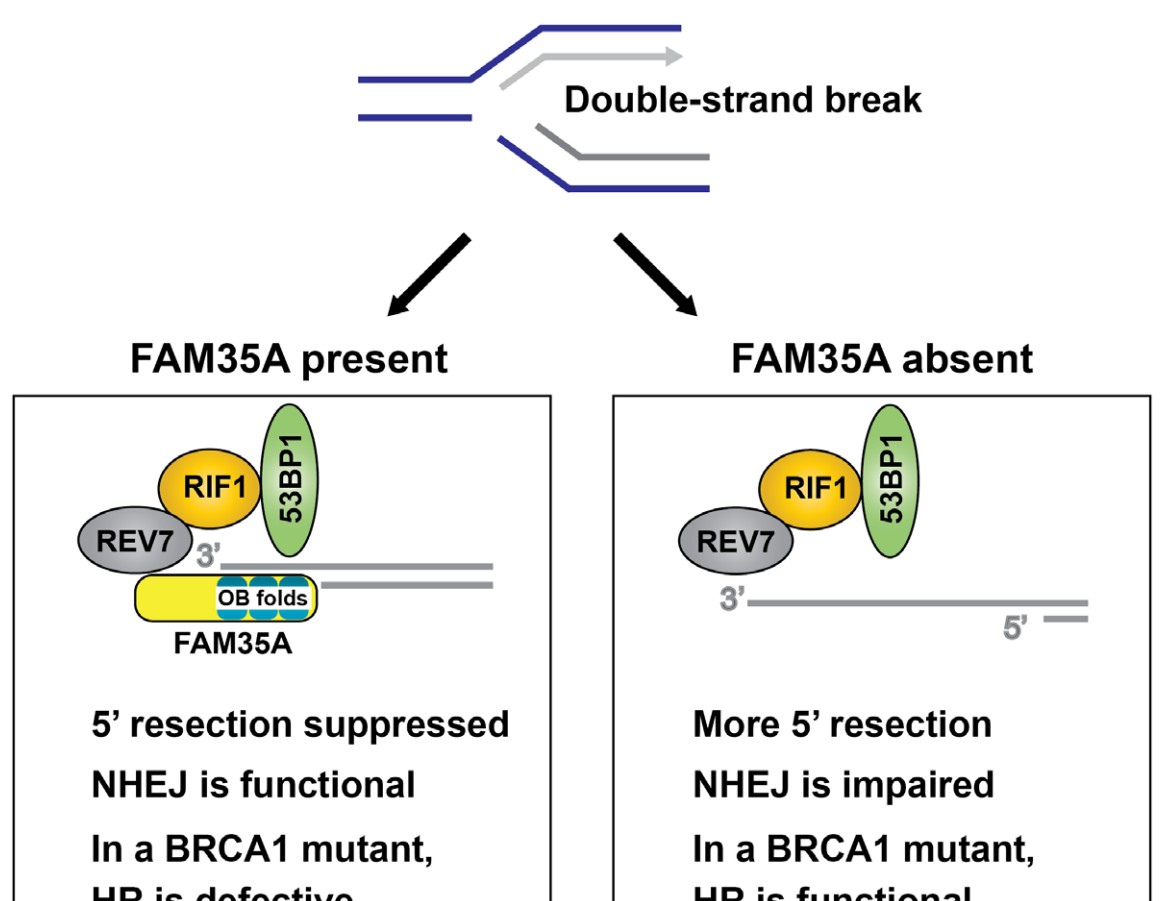

**Figure 7.  Influence of FAM35A on end resection.**

During DNA replication, exposure to camptothecin or PARP inhibitors can block replication forks, sometimes producing a one-ended DSB. FAM35A, in concert with REV7, RIF1, 53BP1, and other proteins, is proposed to be involved in suppressing the resection at a DSB that produces 3′ single-stranded DNA. Cells lacking BRCA1 are unable to counter the inhibition mediated by this protein complex. In FAM35A-defective cells, 5′ end resection can occur even in BRCA1 mutants, activating homologous recombination. This may be one mechanism of acquiring resistance to PARP inhibitor in BRCA1-mutant cells. However, excessive end resection can block the action of NHEJ. As an OB-fold protein, FAM35A may interact with single-stranded DNA and then block nucleases or interactions with other proteins.

  

by DNA polymerase θ-mediated alternative end-joining (Wyatt *et al*, 2016).

### FAM35A alterations and implications for cancers

Currently, there are two major models to account for PARP inhibitor resistance in BRCA1-mutant cells: (i) 5′ resection through dysfunction of 53BP1, RIF1, or REV7 (Boersma *et al*, 2015; Xu *et al*, 2015) and (ii) deletion of recruiters (PTIP, CHD4, and PARP1) of enzymes that degrade newly synthesized DNA (Chaudhuri *et al*, 2016). Our discovery of FAM35A deletion in HCC1937 alongside reported evidence suggests a plausible mechanism of FAM35A-associated PARP inhibitor resistance in line with the first model, via 5′ resection.

We investigated the incidence of FAM35A alterations in cancer using cancer genomics databases. Mutations and copy number changes are found in FAM35A at a relatively low level (< 5% of cases) in most cancer types. Strikingly, however, prostate cancer (PCa) data show deletion of *FAM35A* in a high fraction (6–13%) of cases (Fig 6A). Prostate cancers are the only cancer type with this level of FAM35A deletion (for cancers with at least 40 cases per dataset). The *PTEN* gene, frequently deleted in PCa, is located on 10q23.2 about 0.5 Mb distal of *FAM35A* (Fig 3B). Some cases of FAM35A deletion are independent of *PTEN* deletion (Fig 6B). A meta-analysis of six datasets containing *FAM35A* mRNA expression data was performed to compare metastatic and primary tumors in prostate cancer. *FAM35A* mRNA expression is significantly reduced in metastatic prostate cancer (Fig 6C). In prostate cancers, deletion of FAM35A is essentially exclusive from mutation or deletion of BRCA1. From the data presented here, tumors with FAM35A deletion might be predicted to be susceptible to treatment with DNA cross-linking or double-strand break inducing agents, but not to olaparib. Finally, we note that some other commonly used cancer cell lines harbor genetic deletion for FAM35A as annotated in the CCLE, although each must be confirmed experimentally. We found that prostate cancer cell line LNCaP clone FGC still expresses FAM35A mRNA (Fig EV3B).

The data presented here strongly implicate FAM35A in inhibition of DNA end resection following strand breakage (Fig 7). In BRCA1-mutant cells, knockout of FAM35A allows recombination and/or alternative end-joining repair pathways to be utilized more efficiently and helps explain the relative resistance to PARP inhibitors. Recently, a comprehensive survey of proteins associated with 53BP1 (Gupta *et al*, 2018) uncovered REV7-binding proteins including FAM35A (designated RINN2). siRNA-mediated suppression of BRCA1 in human U2OS cells increased their sensitivity to olaparib. When FAM35A/RINN2 was suppressed in these BRCA1-defective cells, olaparib sensitivity was alleviated. This result is consistent with our interpretation outlined here, from analysis of FAM35A suppression in human cells and absence of expression in PARP inhibitor-resistant HCC1937 tumor cells. Based on the protein interactions uncovered here and our prediction of single-stranded DNA-binding activity, a mechanism of action of FAM35A can be proposed whereby the protein interacts directly with single-stranded DNA via its OB-fold domains. In this way, FAM35A may directly block activity of resection enzymes (BLM-DNA2, BLM-EXO1, and MRE11) or associate with BLM to prevent the interaction of BLM with DNA2 or EXO1.

## Materials and Methods

### Human cell cultures and transfections

HEK293 (ATCC CRL-1573), U2OS (ATCC HTB-96), HEK293T (ATCC CRL-11268), MDA-MB-436 (ATCC HTB-130), HCC1937 (ATCC CRL-2336), and HCC1937 BL (ATCC CRL-2337) cells were maintained in Dulbecco's modified Eagle's medium GlutaMAX$^{TM}$ (Life Technologies, Carlsbad, CA) supplemented with 10% fetal bovine serum and penicillin/streptomycin in a 5% $CO_2$ incubator at 37°C. Human HeLa S3 cells were maintained in RPMI supplemented with 10% fetal bovine serum and penicillin/streptomycin in a 5% $CO_2$ incubator at 37°C. All cell lines were routinely checked for mycoplasma contamination using the MycoAlert detection kit (Lonza). Cell lines were validated by STR DNA fingerprinting by the Cell Line Identification Core of the MD Anderson Cancer Center. The STR profiles were compared to known ATCC fingerprints (ATCC.org), to the Cell Line Integrated Molecular Authentication (CLIMA) database version 0.1.200808 (http://bioinformatics.hsanmartino.it/clima/; Romano *et al*, 2009) and to the MD Anderson cell line fingerprint database.

### siRNA transfection

siRNA transfections were performed with Lipofectamine RNAiMAX (Life Technologies) according to manufacturer's instructions and as described previously (Takata *et al*, 2013; Tomida *et al*, 2013). Messenger RNA target sequences used for siRNAs were as follows: FANCA-specific RNAs (5′-AAGGGUCAAGAGGGAAAAAUA-3′) (Invitrogen), FANCD2-specific RNAs (5′-AAUGAACGCUCUUUAGCAGAC AUGG-3′) (Invitrogen), FAM35A-specific RNA (5′-AAGGAGUGGUUC UGAUUAA-3′) #6 (5′-GUACUAAGAGUUGUUGAUU -3′), #7 (5′-GAA CAGGAUCUACAAACAA-3′), and #8 (5′-GCUCACAGUUCUCUGA AGA-3′) (Thermo Scientific). ON-TARGETplus Non-Targeting siRNAs were used as a negative control designated "siControl" (Thermo Scientific). The siRNAs were introduced into HEK293 cells.

### Immunoprecipitation assay and protein identification

Immunoprecipitation was performed as described (Takata *et al*, 2013; Tomida *et al*, 2013). Forty-eight hours after transfection (for FH-FAM35A and GFP, GFP-53BP1, or GFP-RIF1) in 293T cells, the cells were harvested. Each cell pellet was suspended with 300 μl of 0.5B (500 mM KCl, 20 mM Tris–HCl [pH 8.0], 5 mM MgCl$_2$, 10% glycerol, 1 mM PMSF, 0.1% Tween-20, 10 mM 2-mercaptoethanol), frozen in liquid nitrogen, thawed on ice, and sonicated. After centrifugation, 900 μl of 2B (40 mM Tris–HCl [pH 8.0], 20% glycerol, 0.4 mM EDTA, 0.2% Tween-20) was added to the supernatant and incubated with 10 μl of GFP agarose (MBL) for 4 h at 4°C. The bound proteins were washed with 700 μl of 0.1B (100 mM KCl, 20 mM Tris–HCl [pH 8.0], 5 mM MgCl$_2$, 10% glycerol, 1 mM PMSF, 0.1% Tween-20, 10 mM 2-mercaptoethanol), three times and eluted with 30 μl of 2× SDS loading buffer (100 mM Tris–HCl [pH 6.8], 4% SDS, 0.2% bromophenol blue, 20% glycerol, 200 mM DTT). These samples were separated by polyacrylamide gel electrophoresis, transferred to a membrane, and detected with the indicated antibodies and ECL reagents (GE Healthcare). Protein identification was performed by LC-MS/MS mass spectrometry of gel slices as described previously (Takata *et al*, 2013). Protein identifications were checked for agreement with the molecular

mass predicted from the relevant gel slice. Protein identification established > 99.9% protein probability assigned by the Protein Prophet algorithm, with a minimum of two peptides at 95% peptide probability. Abundant proteins found commonly in immunoprecipitation experiments with these epitope tags and agarose were eliminated from consideration (Takata *et al*, 2013). These included lamins, importins, chaperones, and ribosomal proteins.

## Random plasmid integration assay

The plasmid integration assay was performed as described (Lou *et al*, 2004; Galanty *et al*, 2012; Boersma *et al*, 2015). Plasmid transfections were performed with Polyethylenimine 25kD linear from Polysciences (cat# 23966-2) as described (Lee *et al*, 2014). Briefly, 24 h after transfection (with linearized pDsRed-Monomer-Hyg-C1 (Clontech; #632495) with SmaI or XhoI) in FAM35A-depleted 293T cell lines and control cells, $1.5 \times 10^5$ cells were plated into replicate 10-cm dishes for Hygromycin B treatment and $5 \times 10^2$ cells were plated into replicate 10-cm dishes for plating efficiencies. The next day, Hygromycin B 300 μg/ml was added to the dish, and the cells were incubated until the appearance of colonies (10–14 days). Cells were then fixed, stained, and counted. Statistical analysis was performed by unpaired *t*-test ($P < 0.05$).

## shRNA vectors

shRNA vectors were purchased from MD Anderson core facility: shFAM35A#1 (V2LH2S_221094), shFAM35A#2 (V2LH2S_277883), shFAM35A#3 (V3LH2S_359959), shFAM35A#4 (V3LH2S_359963), and shScramble (RHS4346).

## Antibodies

Antibodies purchased from Sigma-Aldrich, with dilutions for immunoblotting, were as follows: F3165, monoclonal anti-FLAG 1:10,000; T5168, monoclonal anti-α tubulin 1:8,000; A0168 HRP (horseradish peroxidase)-conjugated anti-mouse IgG 1:10,000, A0545 HRP-conjugated anti-rabbit IgG 1:10,000; HPA036582, polyclonal anti-FAM35A 1:200. D153-8 agarose-conjugated anti-GFP (RQ2) was purchased from MBL International Corporation. From Clontech, polyclonal anti-GFP antibody 632592 was used at 1:200 (for immunostaining); monoclonal anti-GFP 632381, 1:3,000. Anti-RAD51 antibody (B01P, 1:2,000) was purchased from Abnova. Goat anti-mouse IgG (H+L) secondary antibody, Alexa Fluor® 594 conjugate (A-11005) 1:2,000, goat anti-rabbit IgG (H+L) secondary antibody, Alexa Fluor® 488 conjugate (A-11008) 1:2,000, and goat anti-mouse IgG (H+L) secondary antibody, Alexa Fluor® 488 conjugate (A-11001) 1:2,000 were purchased from Thermo Fisher. Anti-FANCD2 (EPR2302, 1:2,000 dilution) was purchased from GeneTex, Inc. From Cell Signaling Technology, monoclonal anti-HA (C29F4) was used at 1:1,000.

## DNA constructs

Human *FAM35A* cDNA (Clone ID: 6146593) was obtained from Thermo Scientific. Isoform 2 cDNA was PCR amplified from FAM35A cDNA as a XhoI-NotI fragment with 5′FAM35A (XhoI) primer (5′-CCGCTCGAGATGAGTGGAGGATCTCAAGTCCAC) and 3′FAM35A (NotI) primer (5′-TAAAAGCGGCCGCTCAGAGACGGGCATTGGCTC

CATGC) to clone into pETDuet-1 and pRSFDuet-1 (Novagen). Full-length REV7 cDNA was cloned into pETDuet-1 (Tomida *et al*, 2015). The XhoI-NotI fragments from FAM35A and REV7 were inserted into pOZN and pOZC (Nakatani & Ogryzko, 2003; kindly provided by Hank Heng Qi, Children's Hospital, Boston) and pCDH-FH (Tomida *et al*, 2015). Flag-HA-mouse FAM35A isoform 1 expression vector was PCR amplified from FAM35A cDNA as a XhoI-NotI fragment with 5′mFAM35A (XhoI) primer (5′-CCGCTCGAGATGAGTCAAGG ATCACAAGTTCAC) and 3′mFAM35A (NotI) primer (5′-TAAAA GCGGCCGCTCACTCCTTCCCAGGAATGCCTCC) to clone into pCDH-FH. After construction, expression vectors were confirmed by DNA sequencing. pDESTpcDNA5-FRT/TO-eGFP-Rif1 and pcDNA5-FRT/TO-eGFP-53BP1 were gifts from Daniel Durocher (Addgene plasmid # 52506 and # 60813, respectively; Escribano-Diaz *et al*, 2013; Fradet-Turcotte *et al*, 2013).

## DNA damage sensitivity in FAM35A-depleted cell lines

To analyze sensitivity to chemical DNA-damaging agents, HEK293 were plated into white 96-well plates (1,250 cells/well). The next day, various concentrations of MMC, etoposide, and olaparib were added to the wells, and the cells were incubated for 48 h. The cells were lysed and a reagent that luminesces in the presence of ATP was added (ATPLite One Step, Perkin Elmer). Luminescence was measured using a plate reader (Biotek Synergy II) and normalized to undamaged control. For the clonogenic assay, $1.5 \times 10^5$ cells were plated in 10-mm culture plates and incubated for 24 h prior to DNA damage induction. Groups of plates were exposed to camptothecin (20 nM). After 24 h, medium was changed, and cells were incubated until the appearance of colonies (14–21 days). Cells were then fixed, stained, and counted. Statistical analysis was performed by unpaired *t*-test ($P < 0.05$).

## Foci analysis by microscopy

Immunofluorescence photography used a Leica DMI6000B microscope as described previously (Takata *et al*, 2013). U2OS cells stably expressing GFP-FAM35A were plated into an 8-well chamber slide. Cells were exposed to MMC (100 ng/ml, 24 h) and were fixed the following day with paraformaldehyde and stained for DAPI and GFP (Tomida *et al*, 2013). To measure the formation of DNA double-strand breaks and downstream signaling, FAM35A acutely depleted and control HEK293 cells were plated in 8-well chamber slides, treated with MMC (100 ng/ml 24 h), fixed with paraformaldehyde, and stained for DAPI and γH2AX or FANCD2 (Song *et al*, 2010). γH2AX or FANCD2 foci numbers were counted and plotted. FAM35A-depleted BRCA1-mutant cells were plated in 4-well chamber slides and exposed to X-rays the following day (10 Gy, 2 Gy/min, 160 kV peak energy with a Rad Source 2000 irradiator, Suwanee, GA). Six hours later, cells were fixed with paraformaldehyde and stained for DAPI and RAD51 (Song *et al*, 2010; Johnson *et al*, 2013).

## Genomic PCR

Total DNA was isolated from HCC1937 and MDA-MB-436 cells using a DNeasy Blood & Tissue Kit (Qiagen) following the manufacturer's protocol. Total DNA (400 ng) was used as a template for genomic DNA amplification of FAM35A exons 1 and 2 by PCR with primer 1

(5′-ACGGGCGGCCGGATTTGCCCGGAGG) and primer 2 (5′-CTTAA TCTTTTGACCATCAAGG). Control primers were REV3L-F (5′-ACAG CTTCAGAGGAAAGCCA) and REV3L-R (5′-GTTACCGATCGTGT CCGTTT).

### Quantitative PCR (qPCR) assay

Total RNA was extracted using an RNeasy Mini Kit (Qiagen) following the manufacturer's protocol including on-column DNase treatment. Total RNA (1 μg) was used as a template to synthesize cDNA with the High-Capacity cDNA Reverse Transcription Kit (Thermo Fisher). qPCR was then performed on the ABI 7900HT Fast Real-Time PCR System (Applied Biosystems). TaqMan primers and the probe sets for *hFAM35A* were purchased from Thermo Fisher (assay ID Hs04189036_m1). The absolute quantity (AQ) of transcripts for *hFAM35A* was determined using the generated standard curves. Standard curves for each gene were determined using the plasmid pRSFDuet-1 carrying one of *hFAM35A*. Each reaction was performed in triplicate.

### Bioinformatics

Analysis of predicted protein disorder used the DisEMBL (http://dis.embl.de), Pondr (http://pondr.com), and Phyre2 web-based servers (Kelley *et al*, 2015). Protein structure prediction was cross-checked using multiple tools. Several structure prediction servers identified models corresponding to OB domain A. ModWeb (https://modbase.compbio.ucsf.edu) predicts high confidence similarity (sequence identify 18%) of FAM35A residues corresponding to OB fold A with PDB code 2K50 (*Methanobacterium thermoautotrophicum* RPA protein OB fold) and other with other RPA models. SwissModel (https://swissmodel.expasy.org/) identified homology of FAM35A 904 aa isoform residues 583-735 (OB fold B) with human RPA1 OB-fold PDB code 1FGU. When the C-terminal 480 amino acids of FAM35A were submitted to Phyre2 for structure prediction, the highest scoring template was PDB code 4GOP (71% coverage, 99.4% confidence). OB folds A, B, and the first part of OB folds C were modeled from 4GOP. The OB fold C model and is consistent with a partial OB fold predicted for FAM35A residues 736-836, PDB code 3U50.

Cancer genome data and Cancer Cell Line Encyclopedia data were accessed from the cBioPortal (www.cbioportal.org) for Cancer Genomics (Gao *et al*, 2013). Oncomine (Rhodes *et al*, 2007) (www.oncomine.com) was used for meta-analysis of six datasets containing FAM35A mRNA expression data (all six datasets are microarrays) with comparisons between metastatic and primary tumor in prostate cancer. Total patient numbers and detailed information regarding published datasets and associated publications are indicated in Fig 4B.

**Expanded View** for this article is available online.

### Acknowledgments

We thank Edmund O'Brien, Sara K. Martin, Megan Lowery, and Karen Boulware for assistance with experiments, Mary Walker for editorial help, Dr. Stefan Arold for initial OB-fold structural predictions in 2012, and Dr. Sylvie Doublié for PONDR disorder predictions and discussion. This research was supported by National Institutes of Health (NIH) grants CA132840, CA097175, CA193124 and the Grady F. Saunders Ph.D. Distinguished Research Professorship (to RDW), grant RP130297 from the Cancer Prevention and Research Institute of Texas (to RDW), and National Institutes of Health (NIH) grants CA212556 (to JT), and The University of Texas MD Anderson Cancer Center Institutional Research Grant and Center for Radiation Oncology Research (to KT). We also acknowledge funding from CPRIT Core Facility Support grant RP120348 and RP170002 and NIH Cancer Center Support Grant P30-CA016672 (University of Texas M. D. Anderson Cancer Center) and CPRIT grant RP110782 (to M.D.P.).

### Author contributions

JT and K-iT conceived and designed the experiments. JT and RDW designed the research. JT, SB, and MDP performed research. JT, K-iT, H-PC, DGT, and RDW analyzed data. JT and RDW wrote the manuscript.

### Conflict of interest

The authors declare that they have no conflict of interest.

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
