## [Review process File · The EMBO Journal]

FAM35A associates with REV7 and modulates DNA damage responses of normal and BRCA1-defective cells

Junya Tomida, Kei-ichi Takata, Sarita Bhetawal, Maria D. Person, Hsueh-Ping Chao, Dean G. Tang and Richard D. Wood

Review timeline:	Submission date:	1 April 2018
	Editorial Decision:	24 April 2018
	Revision received:	3 May 2018
	Accepted:	7 May 2018

Editor: Hartmut Vodermaier

Transaction Report:

1st Editorial Decision

24 April 2018

Thank you for submitting your manuscript on FAM35A as new REV7 partner to The EMBO Journal. We sent it to three expert referees and while two reports are already in, a delayed third one is still outstanding. Given the general agreement of the two reports at hand, and to avoid unnecessary loss of time (also in light of a very recent online publication of a related study by Gupta et al), I decided to forward these comments to you already at the present stage together with an invitation to prepare a revised version of the manuscript.

Key points that this revision should address are the related points 1 and 2 of referee 1 (regarding protein expression levels/endogenous interactions), as well as this referee's point 4 regarding different drug sensitivities. On the other hand, animal or primary cell line studies (pt. 6) as well as in-depth mechanistic follow-up on olaparib sensitivity (pt. 4) would not be necessary at this stage. Furthermore, the manuscript would clearly be strengthened by any data you may be able to provide in order to more directly support FAM35A resection roles (ref 1 pt. 5) or FAM35A ssDNA binding in general (ref 2 pt. 1). Finally, regarding the publication by Gupta et al that appeared only after your submission to our office, it should be appropriate to briefly mention/discuss/compare this work towards the end of the discussion section.

Thank you again for the opportunity to consider this work for The EMBO Journal, and I look forward to your revision!

REFeree REPORTS

Referee #1:

The authors use an immunoprecipitation / mass spectrometry approach to identify FAM35A as a novel interacting partner for REV7. REV7 is implicated in modulating the ability of BRCA1-deficient cells to repair DNA damage. Knockdown studies of FAM35A indicate that it, too, helps

protect cells from the effects of certain DNA-damaging drugs. FAM35A can be found in nuclear damage foci when overexpressed. It also modulates the sensitivity of BRCA1-deficient cell lines to the cytotoxic effects of camptothecin. Conserved structural elements in FAM35A are discussed, along with the degree of expression of FAM35A in cancer cases (using pre-existing data sets).

Gupta et al recently published a paper in 'Cell' identifying FAM35A as one of a number of components of a 'shieldin' complex that acts downstream of 53BP1 to modulate drug sensitivity in BRCA1-deficient cells. The Gupta study is substantially more detailed than the Tomida manuscript, although the Tomida manuscript could potentially be considered an interesting companion study. This study technically appeared first, on bioRxiv. The study certainly contains interesting findings, but is quite preliminary and short on mechanistic detail.

Major points.

1. In Figure 1C the authors co-transfect FAM35A along with RIF1 or 53BP1, and note that the exogenous proteins can be co-immunoprecipitated. This experiment is somewhat weak, because the proteins are almost certainly overexpressed.
2. Equivalently, for the experiment using GFP-FAM35A in U2OS cells described in Figure 2A, was the level of FAM35A expression in any way similar to endogenous levels?
3. Depletion of FAM35A in HEK293 cells correlated with sensitivity to MMC and etoposide, but not olaparib (Fig 2C). This is an interesting result, but no mechanism is offered. In particular, why are cells lacking FAM35A sensitive to etoposide but not olaparib?
4. shRNA of FAM35A in BRCA1-deficient MDA-MB-436 cells altered the sensitivity of these cells to camptothecin. This is an interesting result. However, I find myself wondering what the effect was on sensitivity of these cells to MMC, etoposide and olaparib?
5. The authors suggest that FAM35A modulates resection of DNA double-strand breaks, but never attempt to measure resection.
6. The study uses a lot of cell lines, and no primary cells. A more complete study would analyze the phenotypes of a knockout mouse, and do genetics with that.

Minor Points

1. I seem to be missing a figure legend for Supplemental Figure 4.

Referee #3:

Tomida et al. have identified that FAM35A is a protein implicated in the repair of DNA double strand breaks, and have suggested a more precise role in inhibiting DNA end resection. The authors have presented data supporting FAM35A proposed role through a number of techniques, including immunoprecipitation assays, sequence analysis, immunofluorescence imaging and qPCR, and underlines the protein's significance through analysis of cancer genomic databases. The focus of this manuscript is on a novel DNA repair protein that will be of particular interest to the genomic integrity community, as well as appealing to a broader scientific audience. Overall, this is an elegant and important study that utilizes a well thought out experimental design and merits publications in EMBOJ.

Some of the findings here relate to a very recent paper published in Cell (Gupta et al. Cell 2018), and therefore the authors might want to discuss the relevance.

Minor points:

1. This manuscript suggests that FAM35A holds some affinity for ssDNA, and it would also be interesting to see what DNA substrates it prefers to bind to (or if there is any preference) and in what preference, though this might be beyond the scope of this study.
2. Does a sequence/structure analysis provide any clues to which residues are in contact with any of

the proteins listed that are associated with FAM35A?

3. The paper makes claims that FAM35A is a factor as to which DNA repair pathway takes precedence. Can the authors comment as to whether the expression levels or the phosphorylation of FAM35A are cell cycle dependent?

1st Revision - authors' response

3 May 2018

Our responses and action in reply to the comments are given in *italics* below.

Referee #1:

The authors use an immunoprecipitation / mass spectrometry approach to identify FAM35A as a novel interacting partner for REV7. REV7 is implicated in modulating the ability of BRCA1-deficient cells to repair DNA damage. Knockdown studies of FAM35A indicate that it, too, helps protect cells from the effects of certain DNA-damaging drugs. FAM35A can be found in nuclear damage foci when overexpressed. It also modulates the sensitivity of BRCA1-deficient cell lines to the cytotoxic effects of camptothecin. Conserved structural elements in FAM35A are discussed, along with the degree of expression of FAM35A in cancer cases (using pre-existing data sets).

Gupta et al recently published a paper in 'Cell' identifying FAM35A as one of a number of components of a 'shieldin' complex that acts downstream of 53BP1 to modulate drug sensitivity in BRCA1-deficient cells. The Gupta study is substantially more detailed than the Tomida manuscript, although the Tomida manuscript could potentially be considered an interesting companion study. This study technically appeared first, on bioRxiv. The study certainly contains interesting findings, but is quite preliminary and short on mechanistic detail.

Our study represents the first comprehensive analysis of FAM35A, and we agree that it is a prelude to many further studies that will follow. In the sense that we have carefully confirmed each of our results, we do not consider our data as preliminary. As recommended, we now briefly mention relevant aspects of the very recent Gupta study towards the end of the paper on page 9 and in consideration of this have removed the phrase "first analysis" from page 3. The Gupta et al publication reports a comprehensive proteomic survey of protein associations and focuses on a Rev7 binding protein Rinn1. The FAM35A protein was designated RINN2 by those authors. Gupta et al report a few experiments with FAM35A/RINN2: they showed that suppression of the gene in U2OS cells causes a modest sensitivity to ionizing radiation, that suppression does not cause olaparib sensitivity, but that when FAM35A/RINN2 was suppressed in cells with reduced BRCA1, olaparib sensitivity was alleviated. Our paper reports important findings that are not covered at all in the Gupta et al. paper. Amongst these: We analyze the protein sequence to show that FAM35A is an OB fold protein, we identify a human PARP-resistant breast tumor as FAM35A-deleted, and we show that RAD51 is still loaded in a FAM35A mutant.

Major points.

1. In Figure 1C the authors co-transfect FAM35A along with RIF1 or 53BP1, and note that the exogenous proteins can be co-immunoprecipitated. This experiment is somewhat weak, because the proteins are almost certainly overexpressed.

The purpose of the experiment in Fig 1C was to test the detection of RIF1 and 53BP1 from our proteomic screen. In that experiment (Fig 1B), FAM35A was overexpressed in cells and native levels (non-overexpressed) of RIF1 and 53BP1 were detected in the FAM35A immunoprecipitate. As these factors control DNA end-resection, we wanted to go a step further and test a potential association a different way, by co-expressing both proteins with tags as shown in Fig 1C. From this experiment we cannot determine the relative strength of interactions of FAM35A with RIF1 and 53BP1. Our report points this pioneering finding which will certainly be followed up in the future.

2. Equivalently, for the experiment using GFP-FAM35A in U2OS cells described in Figure 2A, was the level of FAM35A expression in any way similar to endogenous levels?

The experiments expressing FAM35A (new Figure 3A) used a vector with a strong promoter, so expression probably substantially exceeds endogenous levels. Overexpression of fluorescently tagged proteins to test relocalization in cells is commonly used in first assessments of DNA damage responses for newly discovered proteins.

3. Depletion of FAM35A in HEK293 cells correlated with sensitivity to MMC and etoposide, but not olaparib (Fig 2C). This is an interesting result, but no mechanism is offered. In particular, why are cells lacking FAM35A sensitive to etoposide but not olaparib?

This is an important point, and the explanation gives some mechanistic insight, and so we have elaborated on it further in the paper, see pages 6 and 7. The most straightforward interpretation is as follows. We suggest that FAM35A is a factor that controls the extent of resection at a double-strand break (DSB). When FAM35A is defective, resection is expected to still occur, probably more extensively than normal, as established previously for RIF1, 53BP1 and REV7. Under conditions of excessive resection, NHEJ is not possible. Loss of this first-choice pathway for repair of a DSB causes some sensitivity to treatment with agents that lead to DSBs (etoposide, MMC). However, homologous recombination should still be intact in FAM35A-defective cells (as it is in RIF1, 53BP1 and REV7-defective cells). Olaparib sensitivity only manifests in homologous recombination defective cells (notably in BRCA1-defective cells). Similar to defects in RIF1, 53BP1 or REV7, a FAM35A defect does not increase olaparib sensitivity in BRCA1-proficient cells.

4. shRNA of FAM35A in BRCA1-deficient MDA-MB-436 cells altered the sensitivity of these cells to camptothecin. This is an interesting result. However, I find myself wondering what the effect was on sensitivity of these cells to MMC, etoposide and olaparib?

Yes, we plan more experiments in this area, but such experiments take considerable time to do properly and we do not have further data to add at present. The recently published Gupta et al study yielded part of the answer, which is that olaparib resistance was increased by FAM35A suppression in BRCA1-defective cells (see that paper's Fig. 7A). We understand that publication of further papers on this subject is imminent.

5. The authors suggest that FAM35A modulates resection of DNA double-strand breaks, but never attempt to measure resection.

This is a research area that will be stimulated by our discovery of FAM35A as a REV7-associated factor. By broadly accessing RAD51 loading, we show that resection is intact in FAM35A-defective HEK293 cells (Figure 3F), is low in a BRCA1-defective cell line, but is restored by FAM35A suppression in the BRCA1-defective cell line (Fig 5B). See the source data for Fig 5B for the best view of this.

As shown in Fig 4C, suppression of FAM35A reduces the frequency of random integration, a readout of NHEJ activity. To strengthen this point, we have carried out the experiments by re-expressing FAM35A isoform 1 in FAM35A defective cells. This data is now added in the revised paper as Figure 4D and E. This demonstrates that the FAM35A defect is responsible for the NHEJ defect, supporting the interpretation.

Quantitative measurement of resection will be important and will require specialized techniques and further time.

6. The study uses a lot of cell lines, and no primary cells. A more complete study would analyze the phenotypes of a knockout mouse, and do genetics with that.

This is absolutely true, that comprehensive studies in the field will include experiments with knockout mice. We understand that some have been generated by other groups already. In future years there will be many research groups crossing FAM35A knockouts with other DNA repair gene mutant mice. It will be possible to generate primary MEFs and ES cells from such mice, which takes care, time and attention in our experience (in addition to substantial project funding).

Minor Points

1. I seem to be missing a figure legend for Supplemental Figure 4.

It is true that there was no Figure Legend, as this simply shows full-gel data for the labeled figures. In the extended view format in this revised version, the full-gel data are linked as figure source data.

Referee #3:

Tomida et al. have identified that FAM35A is a protein implicated in the repair of DNA double strand breaks, and have suggested a more precise role in inhibiting DNA end resection. The authors have presented data supporting FAM35A proposed role through a number of techniques, including immunoprecipitation assays, sequence analysis, immunofluorescence imaging and qPCR, and underlines the protein's significance through analysis of cancer genomic databases. The focus of this manuscript is on a novel DNA repair protein that will be of particular interest to the genomic integrity community, as well as appealing to a broader scientific audience. Overall, this is an elegant and important study that utilizes a well thought out experimental design and merits publications in EMBOJ.

Thank you for these notes, we are also excited about the study.

Some of the findings here relate to a very recent paper published in Cell (Gupta et al. Cell 2018), and therefore the authors might want to discuss the relevance.

Yes, this is now included towards the end of the discussion (see also comments to Referee 1).

Minor points:

1. This manuscript suggests that FAM35A holds some affinity for ssDNA, and it would also be interesting to see what DNA substrates it prefers to bind to (or if there is any preference) and in what preference, though this might be beyond the scope of this study.

This is certainly an important point. We have started producing FAM35A protein in E. coli and baculovirus for such studies. It really merits a careful long-term project. It is probably that the most physiologically relevant study will measure FAM35A binding to DNA in concert with associated factors. We anticipate papers on this in the future!

2. Does a sequence/structure analysis provide any clues to which residues are in contact with any of the proteins listed that are associated with FAM35A?

In theory; the unstructured N-terminal part of the protein is the most likely to interact with other proteins, by analogy with unstructured regions in other proteins. We have added this point briefly, with a literature citation on page 5. REV7 binds to proteins with a consensus of $\phi\phi x PxxxxP$, where ϕ represents an aliphatic amino acid residue (Tomida et al NAR 2015). Such a sequence is not present in FAM35A, but a similar sequence is present in the protein encoded by the RINN1 gene, that was found by Gupta et al (2018) and which associates with FAM35A/RINN2.

3. The paper makes claims that FAM35A is a factor as to which DNA repair pathway takes precedence. Can the authors comment as to whether the expression levels or the phosphorylation of FAM35A are cell cycle dependent?

This is also an interesting point. We have looked for evidence of cell cycle dependence of gene expression in human cells, but information is very sparse. For example the relatively recent genome wide survey in human U2OS cells (Grant et al. 2013 Mol Biol Cell 24, 3634-50, PMID 24109597) used Agilent Whole Human Genome Oligonucleotide arrays (Platform GPL4133). On this array the spot number for FAM35A is A_23_P127279. Querying the relevant data set GSE52100 produces no data for FAM35A. Another aggregate resource is cyclebase.org. There are no cell-cycle related data compiled there for FAM35A. Similarly, no evidence exists yet for cell cycle dependence of phosphorylation of FAM35A. These experiments (and measurements of for the protein expression) will be compelling future research questions.

2nd Editorial Decision

7 May 2018

Thank you for submitting your final revised manuscript for our consideration. I have now had a chance to look through it and to assess your responses to the comments raised by the original reviewers, and found no further objections towards publication. I am therefore pleased to inform you that we have now accepted it for publication in The EMBO Journal.

YOU MUST COMPLETE ALL CELLS WITH A PINK BACKGROUND ↓
PLEASE NOTE THAT THIS CHECKLIST WILL BE PUBLISHED ALONGSIDE YOUR PAPER

Corresponding Author Name: Junya Tomida, Richard D. Wood

Journal Submitted to: The EMBO Journal

Manuscript Number: EMBOJ-2018-99543R